# Preoperative Risk Factors for Periprosthetic Joint Infection: A Narrative Review of the Literature

**DOI:** 10.3390/healthcare12060666

**Published:** 2024-03-15

**Authors:** Ludovico Lucenti, Gianluca Testa, Alessia Caldaci, Fabio Sammartino, Calogero Cicio, Martina Ilardo, Marco Sapienza, Vito Pavone

**Affiliations:** Department of General Surgery and Medical Surgical Specialties, Section of Orthopaedics and Traumatology, Policlinico Rodolico-San Marco, University of Catania, 95123 Catania, Italy; ludovico.lucenti@gmail.com (L.L.); gianpavel@hotmail.com (G.T.); alessia.c.92@hotmail.it (A.C.); sammarfabio@gmail.com (F.S.); ciciocalogero@gmail.com (C.C.); martinailardo52@gmail.com (M.I.); marcosapienza09@yahoo.it (M.S.)

**Keywords:** periprosthetic joint infection, risk factors, PJI, infections, arthroplasty, replacement, prevention

## Abstract

Periprosthetic joint infection (PJI) poses a challenging complication for many patients undergoing arthroplasty, and the literature identifies numerous risk factors. A comprehensive understanding of the primary risk and protective factors for PJI is valuable for surgeons. This article aims to compile and summarize the key risk factors for PJI documented in the literature. Some risk factors are related to the nutritional status of patients, with obesity, weight loss, hypovitaminosis, and malnutrition being frequently reported. Pathologies affecting patients also contribute to PJI risk, including septic arthritis, hepatitis, diabetes, urinary tract infections, anemia, hypothyroidism, osteoporosis, and dental pathologies. Unhealthy habits, such as tobacco and drug abuse, are significant factors. Previous corticosteroid injections may also play a role in infection development. A few protective factors are also reported in the literature (use of statins, preoperative decolonization, and preadmission skin preparation). The identification of risk factors and the implementation of evidence-based preoperative protocols are essential steps in reducing the incidence of PJI.

## 1. Introduction

Periprosthetic joint infection (PJI) remains a substantial and challenging complication that affects many patients, particularly following total knee arthroplasty (TKA) and total hip arthroplasty (THA) [1,2]. The incidence of PJI ranges from 0.2% to 3% [3]. The repercussions on patients are considerable in terms of quality of life, mortality, and hospital readmission, imposing a significant strain on healthcare systems globally [4,5,6,7,8]. PJI causes disability in patients and might necessitate invasive interventions with a potential for substantial adverse events. Precise diagnosis forms the foundational step for initiating effective treatment. The evolution of insights into PJI is an ongoing process, commencing with the refinement of its definition [9] and extending to treatment modalities, encompassing a thorough exploration of primary risk factors.

Regrettably, the scenery of effective strategies to alleviate the burden of this complication remains incompletely charted [10]. Decisions about patient selection, evaluation criteria, comorbidity detection, the quantification of perioperative risk, and the implementation of countermeasures often fall squarely on the shoulders of the treating physician [11].

It becomes increasingly evident that a comprehensive approach is indispensable. This approach should not only acknowledge the specific challenges associated with PJI prevention but also factors in the evolving dynamics of antibiotic resistance [12]. Adequate learning of the main risk and protective factors for PJI can be a good starting point [13]. This article purposes to collect and summarize all the main risk factors for PJI available in the literature. The main aim of this paper is to provide a comprehensive overview of all the recent factors that may influence the risk of PJI. This study has been structured using the most recently published studies on the risk of PJI, to ensure immediate access to up-to-date scientific evidence on this complex and multidisciplinary clinical condition. Major aspects influencing the risk of PJI can be subdivided into risk factors and protective factors, summarized in Table 1 and Figure 1.

Risk factors can be differentiated into several categories: pathologies (diabetes, urinary tract infections, septic arthritis, dental pathologies, anemia, psychiatric disorders, osteoporosis, and hypothyroidism), nutritional status (hypovitaminosis, obesity, malnutrition, and weight loss), substance use (tobacco and drug abuse), and other (season and previous injections). Main defensive factors are related to the use of statins, preoperative decolonization, and preadmission skin preparation.

The incidence of PJI in different joints varies largely, typically falling within the range of 0.5 to 1.0% for hip replacements, 0.5 to 2% for knee replacements, 0.2–26.1% for ankle replacements, and less than 1% for shoulder replacements.

The elevated occurrence of PJI in knee and ankle arthroplasties could potentially be attributed to increased joint mobility and soft tissue vulnerability, coupled with comparatively less protective soft tissue coverage. Specific risk factors for each joint are reported in Table 2.

## 2. Risk Factors

### 2.1. Pathologies

#### 2.1.1. Diabetes

The connection between diabetes and PJI has been largely explored in some studies. These investigations have exposed an odds ratio for surgical site infection and wound complications in diabetic patients between 1.19 and 3.75 [10,30,31] (Figure 2).

Uncontrolled diabetes worsens the clinical status of patients’ outcomes, with an elevated risk of stroke, urinary tract infection, ileus, transfusion, and even mortality [31]. The incidence of diabetes is spiraling upwards, with reports suggesting that 22% of patients undergoing surgery for joint arthritis have diabetes [32,33].

Hemoglobin A1c (HbA1c) screening is one of the most important preoperative assessments needed for all diabetic arthroplasty patients [34]. The revelations from this screening show that 66% of patients slated for TKA are in the spectrum of either diabetic or prediabetic. A further revelation is that 39% of these individuals suffer undiagnosed diabetes or prediabetes [14].

HbA1c and perioperative glucose levels seems to be very important to assess, with studies proposing cutoffs such as 8 mg/dL and 7.7%, correlating with an augmented risk of PJI [35]. Nevertheless, other studies suggest that, while elevated HbA1c is associated with a higher infection risk after total joint arthroplasty (TJA), due to a low sensitivity and reliability, it cannot be considered a solitary indicator of risk for PJI [36,37].

Postoperative blood glucose emerges as a noteworthy glycemic marker associated with PJI. Kheir et al. [38], in their review encompassing over 20,000 arthroplasty patients, observed a positive correlation between PJI incidence and day 1 fasting glucose, with a discernible threshold of 137 mg/dL, irrespective of the patient’s diabetic status. This finding aligns with antecedent studies by Reategui [37] and Mraovic [36], establishing a link between elevated day 1 fasting glucose and an augmented risk of infection. The prevalent practice of reporting morning fasting glucose levels on postoperative day 1 is corroborated. Varady’s exploration [39] into optimal glucose measurement timing revealed a higher prevalence of hyperglycemia on the night of surgery for both diabetic and nondiabetic patients. Moreover, the proposal by Shohat et al. [40] positing glycemic variability as a pivotal parameter associated with PJI and mortality after TJA underscores the complexity of glycemic control in this context. Continued research efforts, including the utilization of continuous glucose monitoring, are imperative to delineate the intricate postoperative glucose profile in arthroplasty patients.

Recent scientific discourse advocates for the adoption of fructosamine as a superior glycemic marker for predicting PJI following TKA [41]. While HbA1c provides a retrospective view of glycemic control over a three-month period, fructosamine offers a more dynamic assessment by reflecting glycated serum proteins with a shorter half-life of 2–3 weeks. A prospective multicenter study involving 1119 patients has indicated a substantial elevation in the risk of PJI, readmission, and reoperation at the three-month postoperative mark for individuals exhibiting elevated fructosamine levels [41]. A proposed threshold for postoperative complications is set at a fructosamine level of 293 µmol/L. Nonetheless, it is imperative to underscore the necessity for further scientific inquiry to elucidate the nuanced roles of diverse glycemic markers and establish their respective thresholds in predicting PJI [41].

According to other authors [13], the preoperative assessment uniformly integrates HbA1c screening for all TJA patients, regardless of their diabetic status. The recent incorporation of fructosamine screening is underpinned by the latest scientific evidence. Noteworthy is the referral of patients with HbA1c levels surpassing 7.5% to an endocrinologist for meticulous optimization before progressing to surgery [13]. Drawing on the authors’ experience, empirical data convincingly demonstrate a significant reduction in the PJI rate among TKA patients undergoing preoperative HbA1c screening and optimization, juxtaposed against historical controls devoid of universal HbA1c screening [13].

#### 2.1.2. Urinary Tract Infections

Urinary tract infections stand as the second most prevalent cause of infection in adults, trailing closely behind respiratory infections. The diagnosis becomes feasible when an observation of 10^5^ CFU/mL aligns with the presence of infection-related signs and symptoms such as fever, pain, leukocytosis, and pyuria. Globally, there are approximately 150 million cases annually, with females being the more commonly affected demographic [42]. Another noteworthy clinical condition is asymptomatic bacteriuria, where bacteria are detected in the urine without concurrent signs or symptoms of infection. While pre-operative urinary tract infection is acknowledged as a risk factor for PJI, the debate lingers on whether preoperative asymptomatic bacteriuria carries a similar risk and whether it warrants antibiotic treatment.

Mayne et al. [42], in their systematic review analyzing three studies, found that the organisms responsible for PJI in patients with preoperative asymptomatic bacteriuria differ from those cultured in their urine during the preoperative period. Consequently, the likelihood of asymptomatic bacteriuria being a risk factor appears diminished. Therefore, the routine antibiotic treatment of asymptomatic bacteriuria in arthroplasty patients is not recommended.

Honkanen et al. [15], in a retrospective study spanning ten years and encompassing all arthroplasty patients treated in a tertiary care hospital (approximately 23,171 joint replacements), scrutinized urine cultures obtained within 90 days before surgery. Their findings challenge the association of asymptomatic bacteriuria with PJI. Notably, they did not identify any cases where PJI resulted from a pathogen identified in preoperative urine culture. Consequently, they argue against the recommendation of preoperative screening and treatment for asymptomatic bacteriuria [15].

#### 2.1.3. Septic Arthritis

The antecedent occurrence of septic arthritis has been duly acknowledged as a noteworthy risk factor for PJI. 

A comprehensive synthesis of aggregated data, drawn from a corpus exceeding 1300 arthroplasties documented in published literature, elucidates a discernible PJI incidence of 5.96% in instances where a precedent infection manifested within the same articular domain. The mitigated risk of infection is ascribed to the temporal dissociation between TKA surgery and the resolution of the antecedent infectious episode [16].

#### 2.1.4. Dental Problems

Dental health could potentially be a risk factor for PJI. A notable percentage, ranging from 4% to 8% of infections, may be attributed to oral flora pathogens [43]. According to other studies, up to 15% of prosthetic joint infections (PJIs) are associated with dental procedures. Controversy exists regarding the role that oral health problems play in the development of PJI [44]. 

Fenske et al. investigated the impact of preoperative oral screening conducted by orthopedic surgeons and dentists, documenting the incidence of PJI in THA and TKA. They proposed a mechanism involving the acute exacerbation of chronic oral inflammation during the early healing phase after TJA. However, no statistically significant relationship was discovered between preoperative oral screening and the early infection rate [17].

Barrere et al.’s systematic review did not identify any studies intended to establish a connection between preoperative dental pathologies and orthopedic infections [43]. In Kwan et al.’s study, acute PJI was diagnosed in 26 (0.30%) and 21 (0.24%) patients who did and did not undergo preoperative dental clearance. Consequently, they inferred that routine dental clearance did not reduce the rate of PJI, and there was no difference in the organism profile between infections after dental clearance and those without dental clearance [45].

Although there is not any concrete proof linking dental infection to joint infection, it seems that patients with risk factors had an increased probability of preoperative evaluations revealing dental infections that prohibited joint replacement surgery compared to patients without risk factors [43].

#### 2.1.5. Anemia

Anemia is recognized as a significant risk factor for PJI in TJA, affecting up to 35% of patients undergoing elective orthopedic surgery. This condition has been associated with an increased risk of perioperative allogenic blood transfusions, which, in turn, is linked to PJI after TJA [46].

Preoperative anemia escalates the risk of PJI, prolongs hospital stays, and increases mortality rates compared to nonanemic patients. Effective management of preoperative anemia is vital to mitigate the risk of PJI and the necessity for blood transfusions. This involves investigating underlying causes during preoperative assessments and implementing appropriate treatments, such as the use of recombinant human erythropoietin and autologous red blood cell transfusions. To reduce PJI incidence, it is recommended to evaluate patients for anemia approximately 30 days before surgery using blood counts, ferritin, vitamin B12, and folic acid. If anemia is detected, patients should receive adequate treatment with iron supplements, vitamin B12, folic acid, or erythropoietin [18].

Conversely, Tella et al. examined PJI incidence following 583 THAs by analyzing hemoglobin levels, white blood cell counts, and blood transfusions in both pre and postoperative phases. They observed no variations between the two cohorts [47].

However, the limited availability of studies and heterogeneous results pose challenges for data analysis [48].

#### 2.1.6. Hypothyroidism

Hypothyroidism is a disease characterized by the reduced production of thyroid hormones, including T3, T4, and calcitonin (which contributes to the regulation of calcium metabolism). The primary functions of hormones T3 and T4 include regulating energy metabolism (carbohydrates, proteins, and fats) and controlling the somatic and psychic development of the individual. These hormones also play roles in regulating body weight, kidney function, heart function, intestinal function, the musculoskeletal system, the immune system, and the production of other hormones. Hypothyroidism is more prevalent in women and is considered a common disease in the Western world [49]. Some authors investigated the correlation between hypothyroidism and PJI.

Tan et al. [19], in their retrospective study analyzing the preoperative comorbidities of 32,289 TJAs performed between 2000 and 2013, identified hypothyroidism as an independent risk factor for PJI. They also observed that patients who developed PJI had higher thyroid-stimulating hormone levels than those without.

Buller et al. [50] conducted a study on Medicare patients undergoing primary TKA between 2005 and 2014. They matched patients with hypothyroidism with age and gender on a 1:1 ratio and compared 90-day postoperative complication rates between matched cohorts. Hypothyroidism was associated with greater odds of postoperative complications, including PJI, compared to matched controls.

These two studies underscore the role of hypothyroidism in the onset of PJI, emphasizing the importance of diagnosing and managing this condition to mitigate infectious risks in patients undergoing prosthetic surgery.

#### 2.1.7. Hepatitis

Hepatitis C is a disease caused by the HCV virus. The World Health Organization (WHO) estimates that approximately 71 million people worldwide are chronic carriers of the HCV virus. The hepatitis C virus is blood-borne, and common infection routes include the use of injection drugs, exposure to unsafe diagnostic or therapeutic procedures, blood transfusions and nonscreened blood products, and sexual practices leading to blood exposure. Sexual or vertical transmission (from an infected mother to a child) is possible but less common. Unfortunately, there is currently no vaccine against HCV. However, antiviral drugs are available for direct action against HCV.

Bendich et al. [20] sought to determine if HCV could be a modifiable risk factor in TJA by comparing postoperative complications among patients with and without preoperative treatment for HCV. They analyzed the US Department of Veterans Affairs dataset of patients undergoing TKA between 2014 and 2018. In their case-control study, HCV-infected patients treated prior to TJA with Direct-Acting Antivirals (DAA) were included in the “treated” group, while HCV-infected patients untreated preoperatively were assigned to the “untreated” group. They found that implant infection rates were statistically lower at 90 days and 1 year postoperatively among HCV-treated patients than among those untreated (Figure 3).

Bedair et al. [51] analyzed a multicenter retrospective database to identify patients diagnosed with HCV who underwent THA between 2006 and 2016. The patients were divided into two groups: treated and untreated. Moreover, treated patients were further subdivided into those receiving interferon or direct antiviral agent therapies. They found that the untreated group had greater surgical complication rates, with a higher rate of PJI. For treated patients, no differences were observed between treatment types for postsurgical complications [51].

Both studies show that treating HCV infection in patients who will undergo prosthetic surgery significantly reduces the risk of PJI.

#### 2.1.8. Osteoporosis

Osteoporosis is one of the most common diseases of our times, affecting more than 200 million people worldwide, and represents a major public health problem, considering that one of the main complications of this disease is fragility fractures, which in turn carry a high social and economic cost [52]. Although multiple pharmacological therapies are available to treat the various stages of this pathology, many patients at risk or already suffering from the disease do not start or do not scrupulously follow the prescribed protocols [52]. The association between osteoporosis and the risk of PJI has not yet been extensively investigated in the literature. A study published by Harris et al. shows that patients with osteoporosis who undergo TKA have a higher risk of undergoing implant revision for PJI in the 5 years following surgery, compared to those with the same age, sex, and comorbidities who did not have osteoporosis [53]. However, further studies are needed to confirm the increased susceptibility to implant infection in this large group of patients.

#### 2.1.9. Psychiatric Disorders

Mood disorders and, in general, psychiatric conditions such as schizophrenia and bipolar disorder also play a role in determining the increased susceptibility to PJI. In fact, it has been shown that patients who have one or more diagnoses of mental illness are exponentially at a greater risk of developing PJI [21]. Concomitant opioid abuse amplifies this risk cumulatively [54].

### 2.2. Substance Use

#### 2.2.1. Tobacco Use

Singh et al. [55] identified a correlation between smoking and an escalated susceptibility to PJI after primary TKA. The tobacco usage status was ascertainable for 95% (n = 7926) of the individuals, with 5% (n = 446) lacking such information; within this cohort, 7% (n = 565) were active tobacco smokers. The hazard ratios for PJI exhibited a discernible elevation in the subset of individuals presently engaged in tobacco use compared to nonusers. Prudent clinical counsel strongly advocates the discontinuation of smoking antecedent to TKA as a proactive measure.

#### 2.2.2. Tobacco and Cannabis Concomitant Use

Epidemiological trends indicate a notable increase in the prevalence of combined tobacco and cannabis users. Using tobacco and cannabis, over a period spanning 90 days to 2 years, exhibited elevated and significantly increased odds of revision at 2 years post-TKA for a PJI other than increased rates of myocardial infarctions, respiratory failures, surgical site infections, and manipulations under anesthesia. The synergistic association between tobacco and cannabis use before primary TKA significantly heightened the risk of PJI from 90 days to 2 years [56].

#### 2.2.3. Drug Abuse

Regarding substance abuse, patients at high risk of developing PJI appear to be opioid abusers (OUDs), especially those who use these substances intravenously. Opioids, in addition to having various adverse effects on the cardiovascular, nervous, and respiratory systems, are known to cause immune system deficiency [57]. Another aspect to be considered in opioid-abusing patients is the possibility of their intravenous use of opioids, which exposes them to an increased risk of bacteremia and thus PJI (Figure 4).

In these cases, it may be advisable to call in a multidisciplinary team and properly assess the risks and benefits of arthroplasty surgery [22]. It has been observed that even excessive opioid use in the early postoperative period (first 3 days), even without a history of previous dependence, is associated with an increased risk of surgical wound dehiscence, PJI, and early return to the hospital, particularly for cumulative opioid intake values of >172 MMEs (morphine milligram equivalents) [27].

### 2.3. Nutritional Status

#### 2.3.1. Vitamin D Deficiency

Vitamin D is recognized for its pivotal role in bone metabolism, functioning notably at the intestinal level. It facilitates the absorption of calcium and phosphorus while reducing the excretion of calcium through urine. Additionally, vitamin D plays a key role in immune system regulation, influencing the activity of T cells, which can replicate and impede external pathogens. Standard values typically fall within the range of 20 to 40 ng/mL. Maintaining vitamin D within this range is crucial to ensure its optimal functionality.

In a systematic review, Kenanidis et al. [58] scrutinized 18 studies encompassing over 8000 knee and 1500 hip arthroplasties. Their findings indicated that individuals with knee and hip PJI exhibited lower vitamin D levels compared to those grappling with aseptic loosening of the implant. Furthermore, they observed that individuals with deficient vitamin D levels might face an elevated risk of postoperative joint stiffness.

Traven et al. [59], in a retrospective review involving approximately 126 revision THA patients, discovered an association between lower vitamin D levels and an increased risk of complications within the initial 90 days, such as PJI, necessitating revision surgery.

Arshi et al. [60] delved into a cost-effectiveness study concerning preoperative vitamin D repletion in TKA. Their analysis encompassed the incidence of vitamin D deficiency, relative complication rates, and the costs associated with serum 25(OH)D repletion and two-stage revision for PJI. Their objective was to evaluate the viability of both selective and nonselective 25(OH)D repletion in primary TKA to prevent PJI. Given the economical nature of vitamin D repletion in contrast to serum laboratory tests, Arshi et al. [60] posit that nonselective repletion of Vitamin D emerges as a more cost-effective approach than selective repletion in the context of preoperative TKA.

#### 2.3.2. Malnutrition

The term malnutrition is defined as a clinical condition characterized by low intakes of protein nutrients contributing to the patient’s frailty. According to the definition, malnutrition is characterized by serum albumin levels below 3.5 g/dL, serum transferrin levels below 200 mg/dL, serum prealbumin levels below 15 gm/dL, and total lymphocyte count (TLC) <1500 cells/mm^3^ [61]. Some authors also incorporate hypovitaminosis, particularly vitamin D deficiency, into the definition, as this deficiency is associated with an increased risk of developing PJI, and its supplementation may decrease the infection risk [60].

It is crucial to emphasize that malnutrition and obesity can coexist in the same patient, and in such cases, the risk of infection can be calculated based on the analysis of the patient’s nutritional deficits through nutritional risk screening (NRS)2. Malnutrition is also associated with postoperative complications following two-stage surgeries performed for PJI, including increased transfusion rates, reinfections, and hospital readmission within 60 days of discharge [62].

Therefore, if nutritional deficiencies, particularly protein and vitamin deficiencies, are identified, appropriate supplementation and correction can decrease the risk of PJI and surgical site infection [63].

#### 2.3.3. Obesity

Currently, it is estimated that 60% of the population in Europe is overweight or obese, according to the latest 2022 WHO report [64]. Obesity is defined as an important risk factor for PJI, pre and postoperatively [64] (Figure 5 and Figure 6).

Patients with a BMI of 35 kg/m^2^ or higher have an up to six times higher risk of prosthetic device infection; therefore, the AAHKS recommends postponing elective prosthetic surgery, particularly for those patients with a BMI of more than 40 kg/m^2^, as they are considered to be at high risk of infection, particularly associated with other comorbidities [65]. Obese patients require more surgical time and longer hospital stays than nonobese patients, factors that contribute to an increased likelihood of PJI [66]. Obesity can also act as a risk factor in synergy with other known risk factors, such as smoking and the presence of inflammatory arthritis, as demonstrated by Xu C. and co.; as a result, the surgeon must carefully assess the patient’s profile from this point of view [67]. Obesity is frequently associated with metabolic syndrome (insulin resistance, abdominal obesity, hypertension, and dyslipidemia), and in individuals with metabolic syndrome and a BMI above 40 kg/m^2^, the risk of undergoing revision and reoperation for PJI is twice as high [23]. Studies show that it is not only high BMI that should be considered as an absolute parameter for calculating the risk of PJI, but it has also been shown that the pathological presence of visceral fat increases the risk of infection [68], as does the amount of adipose tissue located on the skin surface where the incision is made, the latter being a factor, in particular, for TKA [69]. The last two parameters (visceral fat and local adipose tissue) have recently been proposed as objective indicators of PJI risk, independent of BMI value. Recent literature shows that it is preferable to implement a PJI prevention protocol by delaying the intervention until a BMI of less than 40 kg/m^2^ is achieved, although some authors indicate 35 kg/m^2^ as the maximum cut-off, there is control of glycemic values, and education on a correct lifestyle is provided, with the possible help of a nutritionist [18].

#### 2.3.4. Weight Loss

There are few sources in the literature concerning the increased risk of PJI linked to rapid weight loss. The study by Rechenmacher AJ et al. [70] argues that a weight loss of more than 5 per cent of the initial BMI in 6 months may expose one to a greater likelihood of complications than a loss of the same but in 1 year. Further studies will be needed to define a minimum time limit that is safe and does not expose the obese patient to a greater risk of infection.

### 2.4. Other

#### 2.4.1. Injections

The two primary intra-articular medications used to alleviate pain and potentially delay the need for TJA are hyaluronic acid (HA) and corticosteroids (CSs) injections [71]. The risk of infection is a subject of debate and appears to be related to factors such as the temporal gap from TJA, the number of injections, and the type of corticosteroid.

While the exact pathophysiological mechanism underlying the increased risk of infection is unknown, it is likely related to joint contamination during injection and the immunosuppressive properties of these medications. Intra-articular corticosteroids can compromise local and systemic immune defenses by suppressing the hypothalamic–pituitary–adrenal axis for up to 7 days, potentially increasing the survival of bacteria within the joint [24]. Research suggests that the insoluble corticosteroid crystals may become active again following surgery [25].

Moreover, HA joint injections may impair immune function by modifying the synovium, cartilage, and subchondral bone’s ability to produce immunomodulatory factors [24]. Staphylococci and streptococci, according to Makris et al., may produce hyaluronidase and potentially utilize hyaluronan as an energy source and virulence factor, although the exact mechanism remains unknown [72].

Studies often focus on the interval between infiltration and surgery. In the case of knee arthroplasty, Yang et al. conducted a meta-analysis contrasting infection rates in patients with and without intra-articular injection using corticosteroids and HA. The analysis showed higher postoperative infection rates in the injection group, especially within 0–3 months, but the risk considerably decreased after three months. There were no substantial differences between patients receiving HA and corticosteroid injections [24].

Regarding the knee, both Bedard et al. and Cancienne et al. agree that PJI is more likely following intra-articular knee injections before TKA, and the risk is influenced by the time between injection and surgery [73].

On the other hand, Papalia et al. found no correlation between the number and timing of intra-articular injections and the incidence of complications or infections [74].

In the case of hip arthroplasty, Saracco et al. suggest a significantly higher incidence of PJI for patients who had hip injections within three months of THA [75]. However, this association diminishes when THA occurs more than three months after the injection [75]. Similarly, Albanese et al.’s meta-analysis found a significant correlation between THA and PJI after corticosteroid injections, indicating a 20% increased risk for patients receiving intra-articular corticosteroids within three months before TJA [25]. However, other studies reported no discernible rise in the PJI rate due to prior intra-articular corticosteroid injections [76].

Infiltrations of HA within six months before THA were also associated with an elevated risk of infection [71].

The quantity of injections before surgery should also be taken into consideration. Richardson et al. found no difference in the risk of PJI post-TKA between patients receiving multiple versus a single intra-articular injection before surgery [77]. Kokubun et al. and Forlenza et al. did not find differences in infection rates between TKA patients receiving ≤3 vs. >4 injections before surgery [78,79].

Additionally, research is needed to determine the correct relationship between the number of injections received and the risk of PJI.

Lastly, the risk of PJI appears to differ depending on the type of corticosteroid agent used in the preoperative injection within 90 days of arthroplasty. Methylprednisolone acetate and betamethasone, when given within 90 days of primary TKA, were found to associate with a higher risk of infection. However, this risk returned to normal when given more than 90 days before arthroplasty. Triamcinolone or dexamethasone within or beyond 90 days before surgery did not appear to increase the risk of PJI [80].

#### 2.4.2. Season

The continuous change in seasons is characterized by variations in climatic conditions, temperatures, and humidity rates, all of which impact the biology of living organisms. Low winter temperatures predispose individuals to respiratory infections, colds, and pneumonia, while high summer temperatures can lead to increased sweating and hydro electrolyte problems. The heat promotes bacterial proliferation, and continuous sweating, especially in areas with skin wounds like surgical scars, increases the risk of contamination and infection.

Yang et al. [26] conducted a retrospective cohort study to investigate the correlation between seasons and the risk of PJI. They included only patients who experienced PJI within 1 month after TJA. The study revealed that the incidence of PJI in summer is significantly higher than in winter, following both TKA and THA. The authors argue that the summer season independently poses a risk for PJI. Specifically, PJI cases were primarily concentrated in late summer, which was identified as an independent risk factor for PJI.

Considering these findings, surgeons should consider this seasonal risk factor and avoid scheduling prosthetic replacement surgeries during late summer.

#### 2.4.3. Gender and Age

Gender and age were analyzed as nonmodifiable individual factors in the incidence of PJI. Gender shows conflicting results among the selected studies. According to some authors [73], male patients are more vulnerable to PJI than female patients. However, some studies disagreed, stating that women had an elevated risk of deep infection compared to men. Ren et al.’s meta-analysis, which included 19 studies, found that gender had a significant correlation with PJI risk, and female gender may be a protective factor against PJI following THA [81]. Schiffner et al. reported that male gender is related to an increase in the risk of PJI in primary TKA. However, the study does not show an impact of gender on acute and chronic infections and considers a small sample of 57 patients [82].

There has been disagreement among the included studies regarding the effect of age. Due to weakened immunity, older patients are generally more vulnerable to postoperative infections. However, alternative analyses showed that getting older was either a protective factor or had no direct impact on the risk of PJI [81].

#### 2.4.4. Drugs

Inhaled corticosteroids (ICS), vitamin K antagonists (VKA), and amlodipine appear to be associated with the risk of PJI after primary THA. Bruin et al. examined a sample of 5512 patients, of whom 75 developed PJI. Among them, 10 (13%) used ICS perioperatively, 16 (21%) used VKA preoperatively, and 11 (15%) used amlodipine perioperatively [83].

While inhalation is the most common method of administering corticosteroids, most research has concentrated on the oral use of these drugs, but their effect on the immune system is comparable. About 60% of inhaled CS is deposited in the oropharynx, and a lesser amount enters the lungs. It is partly absorbed in the oral cavity and the gastrointestinal tract, from where it reaches the portal circulation and the liver, undergoing first-pass metabolism. The amount of CSI that enters the lungs and the part that is absorbed from the gastrointestinal tract and escapes hepatic metabolism reaches the systemic circulation in an active form.

It is appropriate to take into consideration that these patients have systemic effects from chronic pathologies. Additionally, oxidative and inflammatory frameworks linked to COPD have been shown to increase cytokine production and activity, suggesting that systemic inflammation may increase susceptibility to infections.

Anticoagulants have been described as increasing the risk of developing postoperative wound complications, and according to Parvizi et al. [84], a vitamin K antagonist was associated with a higher risk for PJIs. Persistent wound drainage and hematomas promote bacterial entry, development, and colonization of bacteria in the soft tissue and prosthesis. In most patients, VKAs can be stopped perioperatively, and the international normalized ratio (INR) is monitored, balancing the prevention of thromboembolic complications and the risk of developing a PJI [84].

Regarding amlodipine, in vitro and in vivo animal studies suggest that it might have antibacterial properties inhibiting β-lactamase. Kumar et al. [85] and Yi et al. [86] found that amlodipine inhibits a variety of bacteria both in vitro and in vivo in animal models, but this effect has not been tested in humans. In contrast to previous in vitro and animal studies, Bruin et al. [83] found an increased risk of PJI in patients taking amlodipine. In conclusion, even if a correlation between CSI, VKAs, amlodipine, and the development of PJI is found, it could be determined by the association of the clinical picture and underlying diseases for which these drugs were prescribed.

#### 2.4.5. Preoperatively Elevated Serum Inflammatory Markers

The levels of C-reactive protein (CRP) and erythrocyte sedimentation rate (ESR) are two inflammatory markers that are found to be elevated in multiple pathological conditions, such as inflammatory diseases, high BMI, smoking habits, and both acute and chronic infections. Furthermore, elevated levels of these markers have been observed in patients with osteoarthritis. In a study examining periprosthetic knee infection rates in a sample of 3796 patients who underwent TKA surgery, it was observed that individuals with higher preoperative CRP and ESR values had a greater likelihood of PJI. This association held true regardless of other associated risk factors, such as smoking, obesity, and diabetes [87]. Contrastingly, in another study involving a sample of 351 patients undergoing their first knee implantation by a single surgeon, no statistically significant difference was found between the risk of PJI and preoperative CRP and ESR levels [88]. One reason for this heterogeneity might be that ESR and CRP can be elevated in various conditions, while multiple factors are considered as predisposing the development of PJI independent of inflammation index values. In support of this hypothesis, it was observed that high BMI values are directly correlated with preoperative CRP and ESR values. This correlation could be a potential confounding factor in the preoperative assessment of the risk of infection in patients undergoing elective arthroplasty [28].

## 3. Defensive Factors

### 3.1. Statins

Recent studies support the theory that patients taking statins are at a lower risk of experiencing PJI. In in vitro studies, statins have demonstrated the ability to reduce the capacity of *S. aureus* to produce biofilm [29]. Additionally, statins have exhibited a favorable impact on bone formation and the modulation of proinflammatory cytokines released following prosthetic implant mobilization [89]. Notably, these studies [29,89] were conducted on samples of patients who had undergone hip or knee arthroplasties. Currently, there are no indications for the off-label use of statins as a preventive drug in patients considered to be at risk.

### 3.2. Preoperative Decolonization

The mucous membranes of our body often host bacteria, with Staphylococcus Aureus being the most common in a symbiotic relationship. In specific situations, such as surgical interventions involving synthetic implants or prostheses, a compromised immune system, or particularly stressful periods, infections by opportunistic microorganisms can develop. An ongoing debate in the literature revolves around the effectiveness of carrier screening and bacterial decolonization as tools to reduce the risk of PJI in patients undergoing prosthetic surgery.

Rohrer et al. [90], in their randomized controlled trial with a two-year follow-up, divided patients undergoing prosthetic surgery into two groups: the *S. aureus* carrier group and the noncarrier group based on nasal swab screening results. Both groups were then randomized into intervention and control arms. The “Carrier” group received daily chlorhexidine showers and the application of mupirocin nasal ointment twice a day for 5 days before surgery, while the “No carrier group” was treated with chlorhexidine showers. No instances of PJI were found at the 2-year mark in either the carrier or non-carrier group. Therefore, this study cannot draw a definitive conclusion about the efficacy of preoperative decolonization in reducing PJI.

### 3.3. Preadmission Skin Preparation

It is understood that the entire skin functions as a defense against external pathogens. However, it should be noted that the skin can harbor commensal bacteria, which under normal conditions do not cause any pathology. Nevertheless, when the skin’s integrity, and thus its barrier function, is compromised due to occasional injuries or surgical incisions, these bacteria can enter the circulation and lead to infections. The risk is particularly significant following orthopedic surgery, especially with the implantation of synthetic means or prostheses. The effectiveness of skin disinfection before prosthetic replacement surgery has been a subject of debate in the literature.

Dai et al. [91] conducted a case-control study with a prospective cohort and retrospective controls. The experimental group comprised 1218 patients who used chlorhexidine-impregnated gauze the evening before surgery for skin preparation. This group was compared with a control group of approximately 1033 patients who had undergone TKA four years before this stage. They found that PJI was more frequent in the control group, showing a significant difference. Consequently, they assert that the preadmission use of chlorhexidine-impregnated gauze for skin preparation is an effective practice in reducing the incidence of PJI after TKA procedures.

It is essential to ensure proper skin preparation using chlorhexidine to minimize the risk of infections caused by Cutibacterium acnes, a Gram-positive bacterium that is part of the commensal flora residing on our skin, predominantly inhabiting the hair follicles and sebaceous glands. This is particularly crucial in shoulder surgery [92].

## 4. Conclusions and Future Recommendations

All patients undergoing TJA are at risk for infection and complications. However, various modifiable risk factors have been identified that significantly increase the risk of PJI.
Identifying all risk factors reported in the literature is very challenging. However, it is fundamental to know, understand, and contrast all the factors that may higher the risk of PJI. Optimizing preoperative assessment before elective arthroplasty is crucial for identifying and addressing these diverse risk factors;The absence of widely accepted and standardized protocols for PJI prevention adds complexity to this issue. Although our comprehension of PJI and its associated risks is substantial, it persists as a significant challenge for both arthroplasty surgeons and patients. A pivotal strategy in overcoming this feared complication is prioritizing prevention. The identification of modifiable risk factors and the implementation of evidence-based preoperative protocols becomes paramount, aiming to maximize patient safety and enhance the overall benefits of undergoing arthroplasties. The main limitation is the subjective selection of the articles that was limited to the most recent studies that seemed related to the topic. Furthermore, several factors have been reported without considering the impact they have on the risk of infection and how it differs between individuals or in associations with each other. By evaluating how each factor influences the risk of periprosthetic joint infection, specific preoperative protocols can be created, particularly for patients with multiple comorbidities and more complex clinical conditions.

Although risk factors have been evaluated individually, in clinical practice we are currently dealing with patients with multifactorial clinical cases. Moreover, even considering a single factor, the patient’s response to the arthroplasty may differ for reasons that are not easily identifiable. It could be very useful to adopt a multidisciplinary approach that can provide a comprehensive rating and a wide range of preoperative risk evaluations, employing appropriate assessments and a standardized scoring system that analyze preoperative risk, to take the best preventive measures possible.

## Figures and Tables

**Figure 1 healthcare-12-00666-f001:**
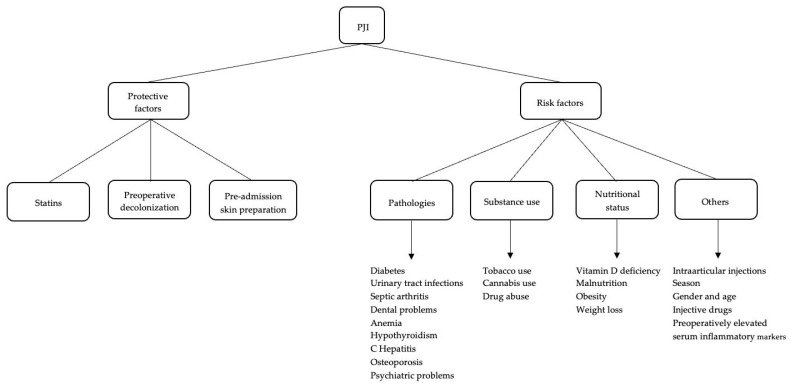
Defensive and risk factors for PJI.

**Figure 2 healthcare-12-00666-f002:**
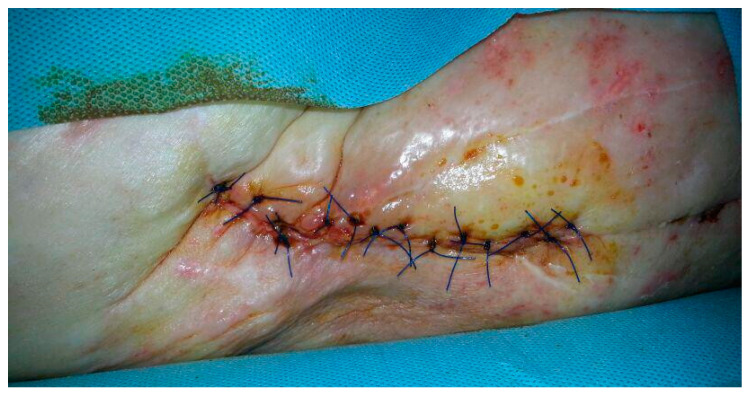
A seventy-nine-year-old male patient with a history of diabetes for around 30 years underwent surgical site infection with discharge on the eleventh postoperative day after a revision of THA after a periprosthetic fracture.

**Figure 3 healthcare-12-00666-f003:**
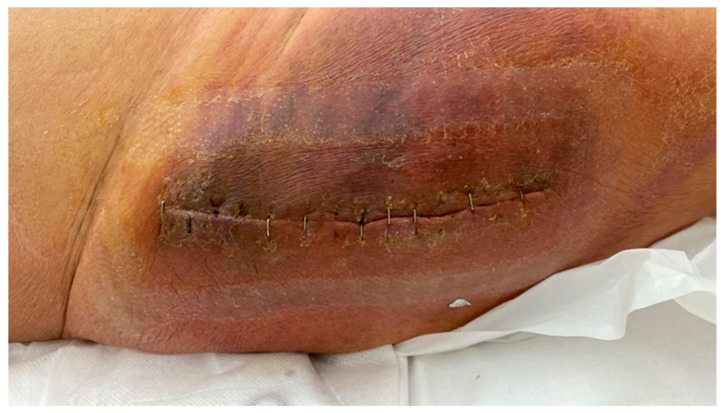
A seventy-seven-year-old female patient with a clinical history of untreated chronic hepatitis C infection and a coagulation disorder presents with extensive hematoma formation and signs of infection. This is a clinical follow-up appointment two weeks after undergoing right hip hemiarthroplasty.

**Figure 4 healthcare-12-00666-f004:**
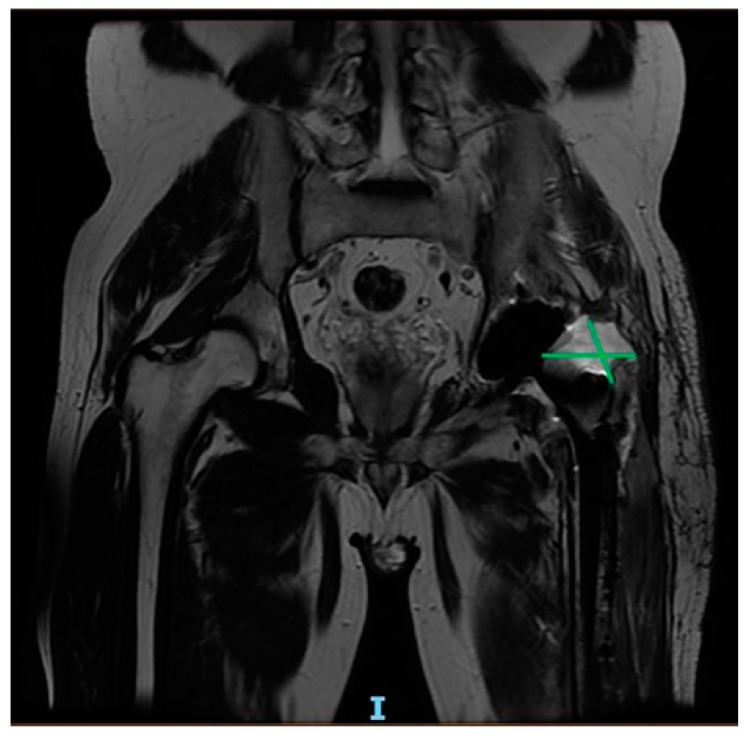
Six-month post-operative MRI scan showing purulent collections in a 64-year-old patient with history of drug abuse, who underwent THA for femoral head fracture.

**Figure 5 healthcare-12-00666-f005:**
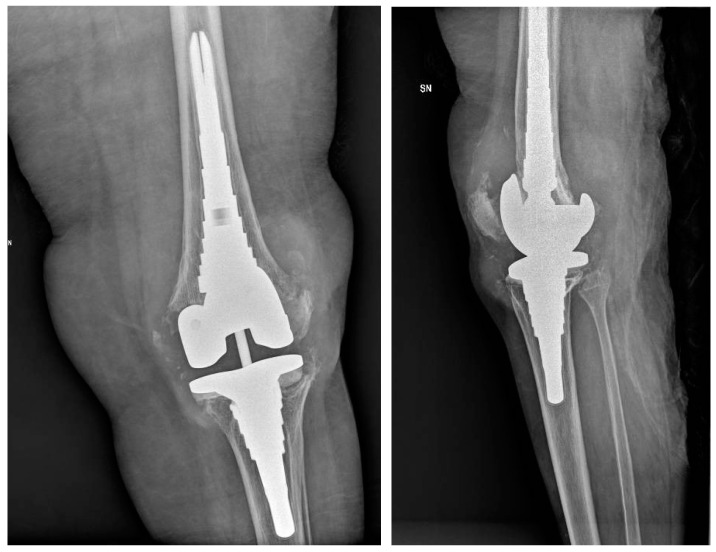
An 86-year-old female patient, obese with a BMI of 32, presents with a history of heart disease, diabetes, and grade II chronic renal failure. Figure 5 shows a left knee revision prosthesis due to a previous periprosthetic infection with signs of osteolysis.

**Figure 6 healthcare-12-00666-f006:**
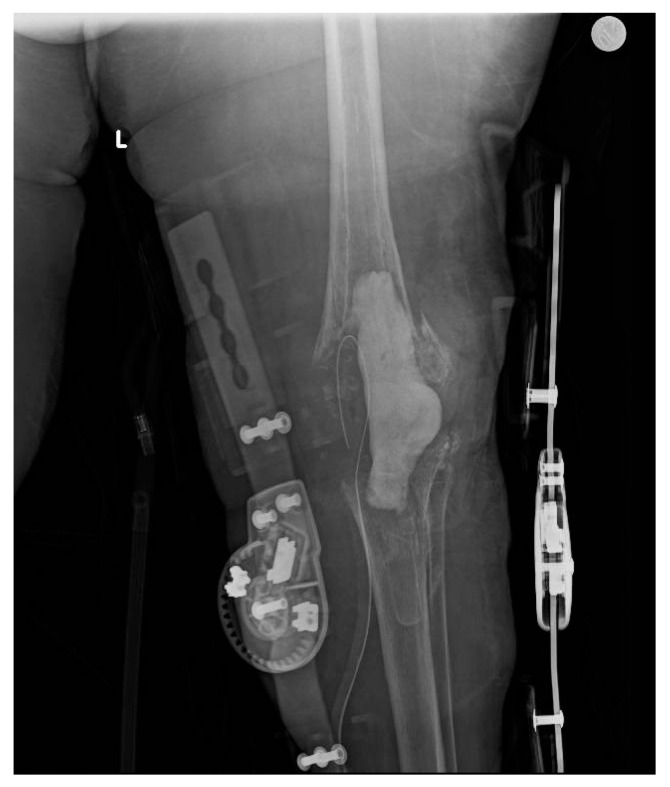
Removal of prosthetic implant, debridement, and placement of antibiotic cement spacer.

**Table 1 healthcare-12-00666-t001:** Risk and defensive factors for PJI. NA, nonavailable (data), TJA: total joint arthroplasty; TKA: total knee arthroplasty; THA: total hip arthroplasty; PJI: periprosthetic joint infection, CRP: C-reactive protein, ESR: Erythrocyte sedimentation rate.

Risk Factors		Prevalence Data
Pathologies	Diabetes	22% of patients undergoing TJA [14]
	Urinary tract infections	from 3.5% to 36% of patients undergoing TJA [15]
	Septic arthritis Dental problems Anemia Hypothyroidism C Hepatitis Osteoporosis Psychiatric disorders	5.96% of patients undergoing TJA [16] 15% of PJI are caused by dental problems [17] up to 35% of patients undergoing TJA [18] 3.4% of patients develop PJI [19] 14.3% untreated patients, 0% treated patients [20] NA 7.7% of patients undergoing TJA [21]
Substance use	Tobacco and cannabis use Drug abuse	NA 2.38% of patients undergoing TJA [22]
Nutritional status	Vitamin D deficiency	NA
	Malnutrition	NA
	Obesity	10% of prevalence in PJI [23]
	Weight loss	NA
Other	Intra-articular injections	from 0.3% to 38.3% according to different studies [24,25]
	Season	1.24% late summer vs. 0.36% in winter [26]
	Gender and age	NA
	Injective drugs	29.3% of patients undergoing TJA [27]
	Preoperatively elevated serum inflammatory markers	12.5% prevalence of PJI with both high CRP and ESR [28]
Protective factors		
Statins		lower incidence at one year (0.36 vs. 0.39%) TKA lower incidence at 90 days (0.37 vs. 0.45%) and 2 years (2% vs. 2.14%) for THA [29]
Preoperative decolonization		NA
Preadmission skin preparation		NA

**Table 2 healthcare-12-00666-t002:** Specific risk factors for each joint.

Joint	Incidence	Risk Factors
Hip	0.5–1%	obesity, malnutrition, hyperglycemia, uncontrolled diabetes mellitus, rheumatoid arthritis, preoperative anemia, cardiovascular disorders, chronic renal failure, smoking, alcohol abuse, and depression
Knee	0.5–2%	male sex, younger age, diabetes mellitus 2, obesity, hypertension, hypoalbuminemia, malnutrition, rheumatoid arthritis, post-traumatic osteoarthritis, preoperative intra-articular injections, previous multiligament knee surgery, previous steroid therapy, tobacco use, procedure type (bilateral), length of stay over 35 days, patellar resurfacing, prolonged operative time, blood transfusions, higher glucose variability in the postoperative phase, congestive heart failure, chronic pulmonary illness, preoperative anemia, depression, renal illness, pulmonary circulation disorders, psychoses, metastatic tumor, peripheral vascular illness, and valvular illness
Ankle	0.2–26.1%	inflammatory arthritis, prior ankle surgery, age < 65 years, BMI < 19, peripheral vascular disease, chronic lung disease, hypothyroidism, low preoperative AOFAS (American Orthopaedic Foot and Ankle Society) hindfoot scores.
Shoulder	<1%	male sex, diabetes mellitus, liver disease, alcohol overuse, iron-deficiency anemia, rheumatoid arthritis, operation history, revision arthroplasty, acute trauma, and nonosteoarthritis as a preoperative diagnosis

## Data Availability

Data are contained within the article.

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
