# Peer review of "Preoperative Risk Factors for Periprosthetic Joint Infection: A Narrative Review of the Literature"

_healthcare, 2024, doi:10.3390/healthcare12060666_

Round 1

Reviewer 1 Report (Previous Reviewer 1)

Comments and Suggestions for Authors

I'm happy with the edits made by the authors. My only concern is that the risk factors were treated individually, where as they may co-occur in clinical situations. The authors note that this is a limitation of the study.

The thesis of this review has also been limited to risk and protective factors, so it's not a complete review. The authors have fleshed out their arguments adequately given this constraint on the scope of the study.

Comments on the Quality of English Language

Minor proof reading is required.

Author Response

Reviewer 2 Report (Previous Reviewer 3)

Comments and Suggestions for Authors

The Authors completed all revisions. 

Author Response

Reviewer 3 Report (New Reviewer)

Comments and Suggestions for Authors

The manuscript "Preoperative risk factors for periprosthetic joint infection: a narrative review of the literature" is aimed to collect and summarize all the main risk factors for PJI available in the literature and to provide a comprehensive overview of all the recent factors that may influence the risk of PJI. The review is well presented. The following comments needs to be addressed prior to consideration for publication.

1. It is rather surprising to see just two images in the entire review. Suggest to include relevant images. This will enhance the worthiness the manuscript.

2. Suggest to include a tree diagram/flow chart indicating all the risk factors, defensive factors.

3. A statistical analysis section is suggested to be included.

4. Please rewrite the conclusions in point-wise manner.

Comments on the Quality of English Language

The general standard of English used is good. please check for occasional spell errors and sentence construct errors.

Author Response

Reviewer 4 Report (New Reviewer)

Comments and Suggestions for Authors

Dear authors,

In this review you sought to summarise the risk factors for periprosthetic joint infection.

Please find my comments below.

Overall, this is a well conducted narrative review. However, I believe there are some limitations that need to be mentioned in the discussion section. For instance, as a foot and ankle surgeon I would like to read more information on ankle joint replacement infections as you have not mentioned anything about this joint. If you are not planning to amend the paper, then you will need to be more specific as to which joints you are dealing with, and this has to be clearly stated in the abstract as well. Also, the risk factors for indolent infections induced by opportunistic microorganisms like C acnes are different for example in shoulder surgery which also needs to be stated in this article.

Furthermore, please provide the evidence behind platelet rich plasma injections and the associated risks as this modality is sometimes used for degenerative osteoarthritis as well.

Comments on the Quality of English Language

None.

Round 2

Reviewer 4 Report (New Reviewer)

Comments and Suggestions for Authors

Dear authors,

Thank you for providing the revised version of your paper. You have now addressed my original comments and the paper is suitable for publication.

This manuscript is a resubmission of an earlier submission. The following is a list of the peer review reports and author responses from that submission.

Round 1

Reviewer 1 Report

Comments and Suggestions for Authors

Comments:

The authors need to have a section explaining their research design, and justification for the selected publications. 

Table 1 breakdown seems to be very shallow. Even if this is a narrative review, there should atleast be a breakdown of the prevalence of the risk/protective factors as a fraction of the available literature.

I would also recommend that the authors specify a thesis for this review. I'm unsure what knowledge gaps this review aims to address. The conclusions section also feels very generic. I would recommend fleshing this section out by tying it to specific challenges identified from the review.

Comments on the Quality of English Language

Needs proof reading. There are minor terminology mistakes(eg. line 75 should be odds ratio and not odd ratio)

Reviewer 2 Report

Comments and Suggestions for Authors

A well written paper but major lacunae 

1. lack of pictorial representation 

2. recent data of different study may added for literature comparison to get more refined information along with major weakness of these studies need to mention.

Reviewer 3 Report

Comments and Suggestions for Authors

The usage of abbreviation is fine after giving what it stands for. The use of ‘Total Knee Arthroplasty, Total Joint Arthroplasty, Periprosthetic Joint Infection, Total Hip Arthroplasty, etc.’ changed in the manuscript and it should be corrected.  

Different parts of the manuscript seem to be written by different Authors and the writing style changes accordingly. The Authors need to review the manuscript according to the writing style.

The table and the title should be the same. The table includes Tobacco use and Cannabis use in the same title in the text.

Line 75: the sentence ends with many studies and the paragraph shows 3 studies. The sentence should be corrected.

Line 87-89: Citation needed for the sentence.

Line 121-123: Citation needed for the sentence.

Line 145-151: Citation needed for the sentence.

Line 163-167: Citation needed for the sentence.

Line 219: Citation needed for the sentence.

Line 254-260: Citation needed for the sentence.

Line 347: Citation needed for the sentence.

Line 360: Citation needed for the sentence.

Line 373: Citation needed for the sentence.

Line 393: The authors clearly explain two parameters.

Line 402: Citation needed for the sentence.

Line 438: Citation needed for the sentence, what are the studies?

Line 487: Citation needed for the sentence., what are the several studies?

Line 524-531: The Authors wrote the in vitro studies suggest ….. but the references include in vivo animal studies. The Authors should correct the sentence.

Line 559: Citation needed for the sentence, what are these studies?

Comments on the Quality of English Language

The language use is OK.

Reviewer 4 Report

Comments and Suggestions for Authors

Many articles have been published on periprosthetic joint infection (PJI) regarding preoperative factors, postoperative factors, diagnosis, and treatment, including a review published in the New England Journal of Medicine in 2023. This article outlines preoperative risk factors for PJI. Due to the large number of publications in this area, it is difficult to find anything new. Diabetes is textbook knowledge, and many of the risk factors discussed in this manuscript are already recognized as risk factors for PJI and have been published in many review articles. Other factors not mentioned in this manuscript include a history of trauma, rheumatoid arthritis, and cardiovascular disease. On the other hand, the inclusion of osteoporosis, which is not widely accepted despite the fact that PJI is an important issue in arthroplasty, and the recommendation, based on only one article, that arthroplasty should not be performed during the summer months when it is not practical, are also inappropriate. There are also problems with the organization of the paper, including the inclusion of the terms "replacement" and "arthroplasty" and the use of abbreviations other than those first used in the paper. In addition, the fact that there is only one listing of the article is undesirable; PJI is an important topic in arthroplasty, but I recognized that the content of the submitted manuscript does not seem worthy of inclusion in Healthcare.